# A protocol for a systematic review of electronic early warning/track-and-trigger systems (EW/TTS) to predict clinical deterioration: Focus on automated features, technologies, and algorithms

**Sharareh Rostam Niakan Kalhori** [1,2]*, **Thomas M. Deserno**[1], **Mostafa Haghi**[3], **Nagarajan Ganapathy**[1,4]

**1** Peter L. Reichertz Institute for Medical Informatics of TU Braunschweig and Hannover Medical School, Braunschweig, Germany, **2** Health Information Management and Medical Informatics Department, School of Allied Medical Sciences, Tehran University of Medical Sciences, Tehran, Iran, **3** Ubiquitous Computing Laboratory, Department of Computer Science, Konstanz University of Applied Sciences, Konstanz, Germany, **4** Biomedical Informatics Laboratory, Department of Biomedical Engineering, Indian Institute of Technology, Hyderabad, India

* sharareh.niakankalhori@plri.de

## Abstract

### Background

This is a systematic review protocol to identify automated features, applied technologies, and algorithms in the electronic early warning/track and triage system (EW/TTS) developed to predict clinical deterioration (CD).

### Methodology

This study will be conducted using PubMed, Scopus, and Web of Science databases to evaluate the features of EW/TTS in terms of their automated features, technologies, and algorithms. To this end, we will include any English articles reporting an EW/TTS without time limitation. Retrieved records will be independently screened by two authors and relevant data will be extracted from studies and abstracted for further analysis. The included articles will be evaluated independently using the JBI critical appraisal checklist by two researchers.

### Discussion

This study is an effort to address the available automated features in the electronic version of the EW/TTS to shed light on the applied technologies, automated level of systems, and utilized algorithms in order to smooth the road toward the fully automated EW/TTS as one of the potential solutions of prevention CD and its adverse consequences.

**Data Availability Statement:** No datasets were generated or analysed during the current study. All relevant data from this study will be made available upon study completion.

**Funding:** SRNK received funds from Alexander von Humboldt Foundation (AVH). The reference number is 3.4-IRN-1185284-GF-E. The AVH website is https://www.humboldt-foundation.de. The funder has no role in the study design data collection and analysis, decision to publish, or preparation of the manuscript.

**Competing interests:** The authors have declared that no competing interests exist.

**Abbreviations:** EW/TTS, Early warning/track-and-trigger systems; JBI, Joanna Briggs Institute; MEWS, Modified Early Warning Score; EWS, Early Warning Score; PEWS, Paediatric early warning systems; MEWT, Maternal Early Warning System.

## Trial registration

**Systematic review registration:** PROSPERO CRD42022334988.

## Introduction

Clinical deterioration (CD) is the physiological decompensation that occurs when a patient experiences a worsening condition or acute onset of a serious physiological disturbance. A deteriorated patient worsens his clinical state and increases morbidity and organ dysfunction, which incurs care escalation, a protracted hospital stay, or even death [1]. CD dynamically causes other primary and secondary adverse consequences (Fig 1) which can be prevented if timely detected [2]. Hence, healthcare systems are required to detect and control patients' physiological decompensation, particularly in high-risk cases [2]. Recognition of patients affected by CD is a priority for many healthcare facilities. Clinicians may apply vital signs tracking tools to detect CDs in the frame of manual charts, scoring tools, or available electronic systems to detect and predict CDs. In this study, we focus on the electronic version of the early warning/track and triage system (EW/TTS).

In 1997, Morgan and colleagues introduced the manual chart of early warning score (EWS) [3], and Maupin et al. reported that EWS in hospitals halves the rate of emergency code calls [4]. All EWS charts follow the same principle including a given threshold score, triggering a prescribed set of actions intended to escalate patient care [5]. The Royal College of Physicians recommended a standardized national EWS in the United Kingdom for all adult inpatients [6]. Early warning charts face limitations such as subjectivity, inadequate nursing skills, infrequent patient monitoring, poor documentation, and a lack of timely action at the time of CD [7, 8]. These restrictions inspired the care provider team in the hospital to improve scoring charts, switching from manual charts to an electronic early warning system called 'early warning /track and triage system' (EW/TTS) [9–11].

EW/TTS is designed based on monitoring patients' vital signs since 85% of severe adverse events (SAE) happen after abnormal vitals [10]. To monitor and detect physiological clinical deterioration, most EW/TTS combine respiration rate, oxygen saturation level, supplementary

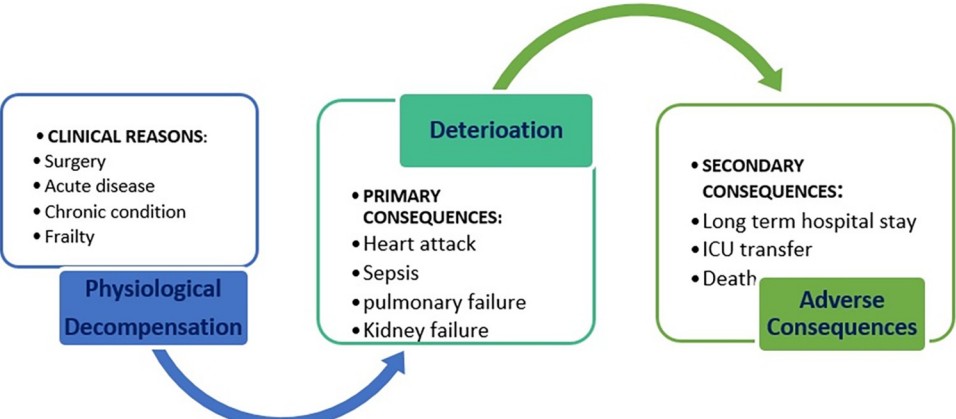

**Fig 1. Dynamic entities of clinical deterioration, primary and secondary consequences in case of improper medical intervention.**

oxygen flow rate, systolic blood pressure, heart rate, temperature, and level of consciousness parameters; some EW/TTSs use the Glasgow coma scale and pain scores too [12]. Staff in emergency departments (ED), ambulatory services, physicians and nurses in ICU and hospital wards [10], and maybe in future residences in smart ambient [13] use EW/TTS. However, the idea of using information technology and artificial intelligence for Electronic Health Records (EHR) connectivity and system automation was brought up by Whittington et al. by developing an automated risk assessment system in hospitals to identify CD-affected patients [14]. Later the idea of EW/TTS improvement was gradually directed toward combating the problems of "failure to identify", "failure to rescue", and "failure to escalate the level of care" for patients at CD risk automatically. Subbe et al. revealed that the deployment of EW/TTS for automated vital signs monitoring and notification at the time of CD is associated with significant improvements in key patient-centered clinical outcomes in hospital wards [15]. By using these systems, it is aimed to activate the rapid response team automatically and speed up the on-time admissions to the intensive care unit (ICU) [15]. Smith et al. conducted a systematic review and concluded the need for electronic system improvement and automation using modern technology to reduce the remaining uncertainty in applying EW/TTS for cardiac arrest and death practically [16].

Currently, many hospital wards and ICUs use patient monitoring tools for CD detection via either an EWS chart or electronic EW/TTS with or without automated features [12]. In addition, the EWS chart has been welcomed in community-based settings such as home, car, and elderly residence dwellings [17]. Also, automatic EW/TTS and embedded alerts are associated with an increase in the accuracy of CD detection from 81 to 100% [10]. Even low-level automated EW/TTS increased the clinical attendance for the patients with the higher early warning score and decreased mortality by approximately 8% in the study period [10]. Due to the high interest to use warning systems and positive signals obtained from applying automated EW/TTS, it seems the next steps are toward full automation EW/TTSs development. However, there might be limitations and challenges in this route. The dynamic entities of CD [1] may cause the current EW/TTS to provide a number of false alerts and then alert fatigue [18]. Furthermore, there are infrequent or delays in vital signs recording which can be resolved by the modern technology of automated capturing of the vital signs using a handheld device. To address solutions for these limitations, studies to apply intelligent algorithms and technologies to support critical decision-making related to CD detection and prediction started on the academic side [19–22], there are, however, so much efforts required to use their results in systems improvement practically. To fulfill this aim, it is required to understand where we stand now to plan for the further step ahead. In the other words, it is expected that applying emerging technologies improve EW/TTS for advanced automated real-time monitoring, timely diagnosis, accurate warning at CD, and precise prediction of SAEs as early as possible. The internet of things (IoT) and artificial intelligence are crucial technologies to bridge the remaining gaps [23, 24]. The full automation of EW/TTS for vital signs monitoring and CD detection and prediction looks ideal, however, it is challenging [25]. Some studies are required to prepare the groundwork for fully automated system development before implementation. The system developers, researchers, users including patients and care providers, policymakers, and vendors need to know about the details of the available electronic EW/TTS in terms of their current situation of automation, applied algorithms, and technologies. This step is important to push them toward further automated EW/TTS.

There are several former reviews focused on the effect of EWS on CD detection [26], the ability of EW/TTS to predict CD [16], machine learning for CD prediction [24, 27], and specific wards in a hospital [28]. However, to plan the future pathway of electronic EW/TTS improvement for more efficient, timely, and accurate CD detection and prediction, we need to understand available automated features supported by technologies, machine learning, and

deep learning algorithms in the artificial intelligence domain. Our study aims to develop a systematically review protocol of automated features, technologies, and algorithms of available EW/TTS for CD detection and prediction in various settings such as hospital wards, smart homes, smart cars, or other possible system deployments.

## Methodology

This systematic review protocol was defined and reported based on the Preferred Reporting Items for Systematic Review and Meta-Analysis Protocols (PRISMA-P) checklist [29]. It is registered in the International Prospective Register of Systematic Reviews (https://www.crd.york.ac.uk/PROSPERO, No.: CRD322838). The PRISMA-P for the protocols checklist is provided as supportive information (S1 Checklist).

### Systematic review objectives and questions

There are nine objectives and questions that will be used to identify the quality of available electronic EW/TTS.

- To identify the automated features of EW/TTS.

- To identify the type of technology applied in EW/TTS.

- To identify the clinical setting (hospital wards or other settings) that EW/TTS is used.

- To identify the clinical parameters or vital signs that are monitored by EW/TTS.

- To identify if EW/TTS monitoring the consequences of CD, including primary and secondary outcomes such as heart failure or death.

- To identify EW/TTS standards (syntax, semantics) for connectivity with other systems such as EHR.

- To identify whether EW/TTS is able to analyze collecting data for CD detection or prediction.

- To identify applied algorithms for data analysis purposes in EW/TTS.

- To identify the evaluation of EW/TTS and its data analytics methods.

### Systematic review questions

- Which automated features does the EW/TTS have?

- What type of technologies in EW/TTS is applied?

- For which settings (hospital wards or other settings) the EW/TTS are used?

- Which clinical parameters or vital signs are monitored by EW/TTS?

- What are the purposes of EW/TTS in terms of the primary and secondary consequences of CD?

- Which standards are used for EW/TTS connectivity?

- Is EW/TTS able to analyze data for CD detection or prediction?

- Which algorithms are used for data analysis in EW/TTS?

- What are the results of evaluating EW/TTS and its algorithms?

## Type of participants

Those studies that reported an electronic EW/TTS with information about the system's ability to monitor clinical parameters, CD occurrence, and primary or secondary consequences such as heart attack, heart failure, or death, possible automation features, the applied technology, setting including hospital wards (ED, ICU, other wards), clinic, or smart ambient, the applied standard for interoperability, the ability of data analysis, the applied algorithms including statistical, machine learning or deep learning methods to predict related outcomes, and the result of evaluating EW/TTS and its data analysis model.

## Information resources

Databases such as PubMed, Web of science, and Google Scholar will be searched for documentation and no restrictions on the type of document will apply. The grey literature must be checked and the two clinical specialists in the area of emergency medicine must approve the sources of search. Initially, the keywords are determined and their synonyms are specified using MESH. Then, English keywords and their combinations will be searched in the aforementioned databases based on the title tag, summary, and keywords without time limitation. The syntax searched in the databases will be as the following section.

## Search strategy

The search study has two main areas of concepts based on the study aims which are set to understand the technical details regarding the EW/TTS. It covers technological terminology AND clinical terminologies. The clinical terms were checked in MESH too. It is aimed to retrieve studies with common themes of the fields of information technology, monitoring systems, and artificial intelligence on one hand, and the fields of triage, emergency medicine, and clinical deterioration on the other hand. we use the following search terms on the title, abstract, and keywords:

( ('patient deterioration') OR ('clinical deterioration') OR ('instability') OR ('failure to rescue') OR ('early warning')OR ('track and trigger') OR ('decompensation') OR ('heart attack') OR ('heart failure') OR ('pulmonary failure') OR ('renal failure') OR ('sepsis') OR ('care escalation') OR ('triage') OR ('emergency') OR ('warn*') ) **AND** ( ('system*') OR ('automat*') OR ('smart*') OR ('wearable*') OR ('internet of thing*') OR ('digital*') OR ('signal*') OR ('sensor*') OR (intelligen*') OR (prediction), (Expert system) ). This general strategy is updated for each database according to its syntax. For example, for search in Scopus, the strategy will be as follows:

(TITLE-ABS-KEY ('patient deterioration') OR TITLE-ABS-KEY ('clinical deterioration')) AND (TITLE-ABS-KEY ('instability') OR TITLE-ABS-KEY ('failure to rescue') OR TITLE-ABS-KEY ('early warning') OR TITLE-ABS-KEY ('track and trigger') OR TITLE-ABS-KEY (decompensation)) OR TITLEABS-KEY ('heart attack') OR TITLEABS-KEY ('heart failure') OR TITLEABS-KEY ('pulmonary failure') OR TITLEABS-KEY ('renal failure') OR TITLEABS-KEY (sepsis)) OR TITLEABS-KEY ('care escalation') OR TITLEABS-KEY (triage) OR TITLEABS-KEY (emergency) OR TITLEABS-KEY (warn*)) AND (TITLE-ABS-KEY (system*) OR TITLE-ABS-KEY (automat*) OR TITLE-ABS-KEY (smart*) OR TITLE-ABS-KEY (wearable*)) OR TITLE-ABS-KEY ('internet of thing*') OR TITLE-ABS-KEY ('digital*') OR TITLE-ABS-KEY ('signal*') OR TITLE-ABS-KEY ('sensor*') OR TITLE-ABS-KEY (intelligen*) OR TITLE-ABS-KEY (prediction) OR TITLE-ABS-KEY (Expert system)).

## Eligibility criteria

### Inclusion criteria

- All original papers are in the English language.

- The works that report electronic systems for CD detection via human vital sign monitoring.

- The studies that report electronic EW/TTS have already been developed and applied in the real world.

- The reported EW/TTSs are applied either for physiological data monitoring or other risk factors of CD detection or monitoring.

- The reported EW/TTS in any setting including hospital, clinic, or smart ambient.

- The reported EW/TTS in every type including bed-connected systems, wearable devices, or unobtrusive sensing technologies.

### Exclusion criteria

- The reports of EW scoring charts manually calculate risk scores such as MEWS, EWS, PEWS, and MEWT.

- All works are published in other languages than English.

- Papers that only provide an abstract.

- Study protocols and ongoing work descriptions.

- Studies not providing any details on technologies, algorithms, or features of the system.

- Studies that only report models or system designs or prototypes.

### Selection process

After searching in the databases, we will transfer the results into an Endnote library and remove duplicates. Then, we will use the Covidence system (https://www.covidence.org) to support the next stages. The titles and summaries will be reviewed to find relevant studies. Subsequently, two experienced scholars specializing in the field of Biomedical Engineering or Medical Informatics will study the full texts independently. We will have group discussions for potential disagreements to resolve the contradictions via online group discussion. A third reviewer will be invited to solve the conflicts in case of further disagreement. To find other related studies, the snowball sampling technique will be used and accordingly, the references cited in the reviewed articles will be taken into account. Highly impact and credible related journals will also be assessed manually to find relevant papers disseminated during the last 10 years. Furthermore, systematic reviews and reference books, and legal documentation will also be checked for related records. Sharareh R. Niakan Kalhori will create an account, particularly for this review, and transfer the Endnote library into Covidence. She will invite other researchers to independently review the records at different levels of study. She will define the criteria and filter in Covidence before starting the study selection.

### Data collection process

After completion of the selection process, the data from each record will be extracted and collected from the full text of the studies. Each study team member will extract data based on pre-

designed variables defined in Covidence. The variables include the content of the 'type of participants' indicated earlier. In case of having difficulty collecting data from the full-text articles, the corresponding author of the study will be contacted through email to obtain the required data. The study's data will be available after completing the review according to the journal policy publishing the final report.

## Data synthesis and statistical analysis

Covidence provides enough options to analyze the results with descriptive parameters such as frequency and percentage, and it supports the generation of graphs. We will present the frequency/percentage of EW/TTS type, the automation feature, the clinical parameters or CD and its consequences to monitor, the applied technologies, algorithms, and evaluation results as available as possible. Furthermore, Covidence exports the spreadsheet of the extracted data and further statistical analysis can be done. There will be different variables such as type of EW/TTS, The group that the system is used for (pediatrics, pregnant women, elderly, and others), the condition or disease that the understudy group has, the monitored clinical parameters or vital signs, the primary and secondary consequences of the monitoring via EW/TTS, the automation features of EW/TTS, the applied technology for automation purposes, the connectivity to the other systems, the applied standards for connectivity, the applied algorithm for data analysis, the results of the algorithm evaluation, and the results of evaluation the EW/TTS. Based on the available variables, the descriptive analysis, including frequency and percentage parameters, will be calculated and presented in the frame of graphs and tables. In the result section, a narrative synthesis will be applied to describe and compare the paper's results. Meta-analysis is not the aim of this systematic review due to the diversity of outcomes and results. Furthermore, correlation analysis between the country of applying EWS/TTS, the system type, the setting and the functionality of the system, the level of automation, applied algorithms, and the results of system or algorithm evaluation will be conducted. Furthermore, we will report the results of the quality assessment of studies in table format.

The results will be provided in the frame of tables, graphs, and numbers with related P-values. We will present the results of our study at relevant conferences and pursue publication in a relevant journal to disseminate findings to serve patients and relevant communities.

## Study quality assessment

The Faculty of Health and Medical Sciences at the University of Adelaide, South Australia has created the JBI critical appraisal checklist (https://joannabriggs.org/), which includes nine main items involving the feasibility, appropriateness, meaningfulness, and effectiveness of healthcare studies [30]. We will use the version for cross-sectional studies.

## Discussion

Artificial intelligence (AI) and related technologies are being increasingly applied to healthcare. These technologies have the potential to change many aspects of patient care due to the complexity and rise of data in healthcare. Several types of AI are already being employed such as automated diagnosis and treatment recommendations, patient engagement and adherence, and administrative activities [31]. Automation is attained from various technologies combination. Machine learning techniques as the most common forms of AI are used for predictive and classification model development through learning from historical training data [18]. The most complex forms of machine learning involve deep learning or neural network models with many levels of features or variables that predict outcomes [31]. Deep learning is also applied to predict CD and other SAE increasingly. Other technologies such as the internet of things (IoT) for patient data

collection through sensing, a different form of natural language processing (NLP) for analyzing unstructured clinical notes on patients, expert systems based on collections of 'if-then' rules for decision-making support, and robots have become more collaborative with humans and are more easily trained by moving them through a desired task [14]. They are also becoming more intelligent as other AI capabilities are being embedded in their 'brains' (operating systems).

System intelligence and automation have a gradual trend and can be improved at different levels from the lowest to the full version. To enhance electronic EW/TTS, in a requirement analysis conducted in 2022, the potential technologies were demonstrated. It is revealed that an intelligent EW/TTS to monitor the patient for CD detection and automatically alarm the rapid response team requires five subsystems that need to be enhanced by proper technologies. They are patient monitoring systems, electronic health records, clinical decision support systems, remote monitoring patients, and dashboards ®istries in an integrated approach using technologies of IoT, deep and machine learning techniques, big data, advanced databases, and standards to create an intelligent EW/TTS [32]. Fulfillment from theory to practice, particularly in the medical area which is a human-related discipline, may face difficulties and challenges. For example, some patients are not keen on the unwieldy vital sign monitoring machine being attached to them continuously. However, they may accept more comfortable and unobtrusive technologies. Other challenges of using AI in critical care such as facets of dependability, reproducibility, and ethics must be considered in the area of automated EW/TTS [33]. There are also benefits of applying an intelligent system for critical care in terms of decision-making aids based on prognosticating the course of the disease and adverse outcomes, patient flow, and hospital beds management, particularly in ICU, capturing complex non-linear relationships, and analysis and representation of unstructured data such as clinical notes [33]. Considering the two sides of the challenges and benefits of applying automated systems, further studies are required to show the justification of intelligent and automated EW/TTS development. The results of this systematic review may trigger a research line toward fully automated EW/TTS development and implementation. For decision-making toward full automation, we need to know where we stand now in terms of applied algorithms, technologies, system connectivity, and level of automation. Further studies are needed to demonstrate the effect of electronic and automated monitoring systems on patient-centric outcomes, and the challenges that may apply in this route. It is necessary to conduct enough studies prior to invest fund, time, and other resources. This review will systematically investigate the available EW/TTS focusing on automated features, technologies, and applied statistical or intelligent algorithms. The result of this study divulges the technical and automation level of current systems. Next generation of automated EW/TTS use the results of this systematic reviews beside applying the timely modern technologies for overcoming the limitations of current the systems.

## Conclusion

Current systems of EW/TTS require further automation and improvement. To understand the current level of automation, applied technologies, and algorithms supporting all beneficiaries and stockholders, this systematic review will be conducted. The results may support developers, policymakers, and vendors moving toward higher-level systems for patient monitoring aimed at the timely detection and prediction of clinical deterioration and its adverse side effects.

## Supporting information

**S1 Checklist. PRISMA-P 2015 checklist.**
(DOCX)

## Acknowledgments

Special thanks to the Alexander von Humboldt Foundation, TU Braunschweig, and Tehran University of Medical Sciences to provide the opportunity for Sharareh R: Niakan Kalhori to conduct this research at PLRI of TU Braunschweig, Germany.

## Author Contributions

**Conceptualization:** Sharareh Rostam Niakan Kalhori, Thomas M. Deserno.

**Data curation:** Sharareh Rostam Niakan Kalhori, Mostafa Haghi, Nagarajan Ganapathy.

**Funding acquisition:** Sharareh Rostam Niakan Kalhori.

**Investigation:** Mostafa Haghi, Nagarajan Ganapathy.

**Methodology:** Sharareh Rostam Niakan Kalhori.

**Supervision:** Thomas M. Deserno.

**Validation:** Mostafa Haghi, Nagarajan Ganapathy.

**Writing – original draft:** Sharareh Rostam Niakan Kalhori.

**Writing – review & editing:** Sharareh Rostam Niakan Kalhori, Thomas M. Deserno.

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
