## [Decision Letter · Decision Letter 0]

6 Sep 2022

PONE-D-22-10855A protocol for a systematic review of early warning/track-and-trigger systems (EW/TTS) to predict clinical deterioration: Focus on automated features, technologies, and algorithmsPLOS ONE

Dear Dr. Sharareh Niakan Kalori,

Thank you for submitting your manuscript to PLOS ONE. After careful consideration, we feel that it has merit but does not fully meet PLOS ONE’s publication criteria as it currently stands. Therefore, we invite you to submit a revised version of the manuscript that addresses the points raised during the review process.

We look forward to receiving your revised manuscript.

Kind regards,

Mohamed Yacin Sikkandar

Academic Editor

PLOS ONE

Journal Requirements:

Additional Editor Comments:

title: Title reflects the scope of the study clearly

Abstract:

1. Too short and it looks like expanded title

2. Methodology, data management and its conclusions can be included

Introduction:

1. Necessity of this study is not clearly explained

Methodology

1. Is there any significance in using the suggested the Search Strategy?

2. Is there any significance in excluding Review articles and conference papers?

3. PICO strategy is used in this study - justification not provided

4. words such as Expert system, prediction could be added in search strategy

5. Technologies and algorithms are not discussed in this paper

General

- manuscript would be incomplete without discussing protocol efficiency 

- though it is a study protocol, challenges and limitations could also be presented

Reviewers' comments:

Reviewer's Responses to Questions

**Comments to the Author**

1. Does the manuscript provide a valid rationale for the proposed study, with clearly identified and justified research questions?

Reviewer #1: No

Reviewer #2: No

2. Is the protocol technically sound and planned in a manner that will lead to a meaningful outcome and allow testing the stated hypotheses?

Reviewer #1: No

Reviewer #2: No

3. Is the methodology feasible and described in sufficient detail to allow the work to be replicable?

Reviewer #1: No

Reviewer #2: No

4. Have the authors described where all data underlying the findings will be made available when the study is complete?

Reviewer #1: No

Reviewer #2: Yes

5. Is the manuscript presented in an intelligible fashion and written in standard English?

Reviewer #1: Yes

Reviewer #2: No

6. Review Comments to the Author

You may also provide optional suggestions and comments to authors that they might find helpful in planning their study.

Reviewer #1: I am unable to understand the significance of this work in its current form.

Authors have merely described some plan of action for performing a review on prediction of clinical deterioration.

Rather, it will be certainly interesting if the review has been actually performed and the findings are discussed in detail.

Reviewer #2: A review to predict clinical deterioration is performed using PRISMA. Following are the few questions to the authors:

1. CD is wide term, authors can try to be more specific

2. Authors have presented few results/observations from earlier publication. It is not clear what critical conclusions are arrived?

3. Authors have used different terms like: Features, AI, IoT etc. But no clear and indepth discussions are found. What is the inference drawn from this? How will it be useful for the future scientists/researchers?

7. PLOS authors have the option to publish the peer review history of their article (what does this mean?). If published, this will include your full peer review and any attached files.

Reviewer #1: No

Reviewer #2: No

---

## [Author Response · Author response to Decision Letter 0]

9 Nov 2022

Dear Editor of PLOS ONE

Thanks so much for the valuable comments on the manuscript (PONE-D-22-10855) entitled '' A protocol for a systematic review of early warning/track-and-trigger systems (EW/TTS) to predict clinical deterioration: Focus on automated features, technologies, and algorithms'' provided by reviewers. On behalf of the team of authors, I am writing the response to those comments which need reaction and correction. The changes are explained here in blue and presented as track changes in the text of the manuscript too. Furthermore, the revised clean version without track changes or highlights is submitted. 

Editor's comments:

1. The abstract is too short and it looks like the expanded title. Methodology, data management, and conclusions can be included.

The abstract is extended with further necessary information. 

2. In the Introduction, the necessity of this study is not clearly explained. 

The necessity of the study is added in the introduction section. 

3. In methodology, is there any significance in using the suggested Search Strategy?

The reason to use the search strategy was added in the Methodology section. The sample for Scopus is added too. 

4. Is there any significance in excluding Review articles and conference papers?

Review articles and conference papers were added to the inclusion criteria. They may have useful information for our study. Thanks 

5. PICO strategy is used in this study, but justification is not provided.

The PICO framework is removed as there is no clinical question based on an intervention here. However, other sections cover the required information regarding the EW/TTS systems, the criteria to compare them, and the possible outcomes considered. 

6. Words such as Expert system, and prediction could be added to the search strategy.

They were added in the related part of the search strategy. 

7. Technologies and algorithms are not discussed in this paper.

They are discussed in the discussion section, Thanks for the comment. 

8. manuscript would be incomplete without discussing protocol efficiency.

The discussion part was improved, thanks for the comment. 

9. Though it is a study protocol, challenges and limitations could also be presented.

challenges and limitations were added in the discussion part. 

Reviews' Comments : 

1. Does the manuscript provide a valid rationale for the proposed study, with clearly identified and justified research questions?

Reviewer #1: No

Reviewer #2: No

The rationale for the proposed study was added in the introduction. Also, the academic problem that will be answered is added more clearly at the end of the introduction. 

2. Is the protocol technically sound and planned in a manner that will lead to a meaningful outcome and allow testing of the stated hypotheses?

The manuscript should describe the methods in sufficient detail to prevent undisclosed flexibility in the experimental procedure or analysis pipeline, including sufficient outcome-neutral conditions (e.g. necessary controls, absence of floor or ceiling effects) to test the proposed hypotheses and a statistical power analysis where applicable. As there may be aspects of the methodology and analysis that can only be refined once the work is undertaken, authors should outline potential assumptions and explicitly describe what aspects of the proposed analyses, if any, are exploratory.

Reviewer #1: No

Reviewer #2: No

The methodology section is extended and more details are added.

3. Is the methodology feasible and described in sufficient detail to allow the work to be replicable?

 Reviewer #1: No

Reviewer #1: No

The methodology section is extended and more details are added.

4. Have the authors described where all data underlying the findings will be made available when the study is complete?

The PLOS Data policy requires authors to make all data underlying the findings described in their manuscript fully available without restriction, with rare exceptions, at the time of publication. The data should be provided as part of the manuscript or its supporting information, or deposited in a public repository. For example, in addition to summary statistics, the data points behind means, medians, and variance measures should be available. If there are restrictions on publicly sharing data—e.g. participant privacy or use of data from a third party—those must be specified.

Reviewer #1: No

Reviewer #2: Yes

We accept to provide the data available after completing the study according to the journal policy that will publish the final report. It is noted in the protocol too. 

5. Is the manuscript presented in an intelligible fashion and written in standard English?

Reviewer #1: Yes

Reviewer #2: No

The whole manuscript was checked for errors and corrections. 

Review Comments to the Author

Reviewer #1: I am unable to understand the significance of this work in its current form.

The authors have merely described some plan of action for performing a review on the prediction of clinical deterioration. Rather, it will be certainly interesting if the review has been actually performed and the findings are discussed in detail.

Reviewer #2: A review to predict clinical deterioration is performed using PRISMA. Following are a few questions for the authors:

1. CD is a wide term; authors can try to be more specific.

A more specific definition is provided in the introduction. 

2. Authors have presented a few results/observations from earlier publications. It is not clear what critical conclusions arrive.

The introduction is improved and a clear conclusion from the review of studies was added. 

3. Authors have used different terms like features, AI, IoT, etc. But no clear and in-depth discussions are found. What is the inference drawn from this? How will it be useful for future scientists/researchers?

The discussion section is added with details regarding the technologies, limitations, challenges, and future works.

---

## [Decision Letter · Decision Letter 1]

29 Nov 2022

PONE-D-22-10855R1A protocol for a systematic review of early warning/track-and-trigger systems (EW/TTS) to predict clinical deterioration: Focus on automated features, technologies, and algorithmsPLOS ONE

Dear Dr. Sharareh Rostam Niakan Kalhori,

Thank you for submitting your manuscript to PLOS ONE. After careful consideration, we feel that it has merit but does not fully meet PLOS ONE’s publication criteria as it currently stands. Therefore, we invite you to submit a revised version of the manuscript that addresses the points raised during the review process.

We look forward to receiving your revised manuscript.

Kind regards,

Mohamed Yacin Sikkandar

Academic Editor

PLOS ONE

Journal Requirements:

Additional Editor Comments (if provided):

Statistical analysis of the outcome or any recommendation to the future researchers is not elaborated.

Reviewers' comments:

Reviewer's Responses to Questions

**Comments to the Author**

1. Does the manuscript provide a valid rationale for the proposed study, with clearly identified and justified research questions?

Reviewer #1: Yes

Reviewer #2: No

2. Is the protocol technically sound and planned in a manner that will lead to a meaningful outcome and allow testing the stated hypotheses?

Reviewer #1: Yes

Reviewer #2: Yes

3. Is the methodology feasible and described in sufficient detail to allow the work to be replicable?

Reviewer #1: Yes

Reviewer #2: Yes

4. Have the authors described where all data underlying the findings will be made available when the study is complete?

Reviewer #1: Yes

Reviewer #2: Yes

5. Is the manuscript presented in an intelligible fashion and written in standard English?

Reviewer #1: Yes

Reviewer #2: Yes

6. Review Comments to the Author

You may also provide optional suggestions and comments to authors that they might find helpful in planning their study.

Reviewer #1: The authors have implemented all the suggestions provided. The manuscript has been now modified with all required details.

Reviewer #2: Yes authors have addressed few of the reviewer's comments. Major portion of the paper talks on general introduction and review process. Statistical analysis of the outcome or any recommendation to the future researchers is not elaborated.

Few grammatical/typos can be looked into carefully.

7. PLOS authors have the option to publish the peer review history of their article (what does this mean?). If published, this will include your full peer review and any attached files.

Reviewer #1: No

Reviewer #2: No

---

## [Author Response · Author response to Decision Letter 1]

21 Dec 2022

Rebuttal Letter Round 2

Dear Editor of PLOS ONE

Thanks so much for the valuable comments on the manuscript (PONE-D-22-10855R1) entitled ''A protocol for a systematic review of early warning/track-and-trigger systems (EW/TTS) to predict clinical deterioration: Focus on automated features, technologies, and algorithms'' provided by reviewers. On behalf of the team of authors, I am writing the response to those comments which need reaction and correction. The changes are explained here in blue and presented as track changes in the text of the manuscript too. Furthermore, the revised clean version without track changes or highlights is submitted. 

Journal Requirements : 

All references were checked. Here there are the changes for the references:

The references 9 and 10 were replaced with each other. But still, we have both of them in the references. 

References 21, 22, 23, 24, and 34 were added. 

2. Statistical analysis of the outcome or any recommendation to future researchers is not elaborated.

The statistical analysis of the outcome and further studies suggestions were added in the discussion section. 

Reviewers' comments:

1. Does the manuscript provide a valid rationale for the proposed study, with clearly identified and justified research questions?

Reviewer #1: Yes

Reviewer #2: No

The introduction section is revised to improve the study justifications. Also, in the discussion section, we added the provided knowledge by this study which can be the basis for further studies and system developments. 

2. Is the protocol technically sound and planned in a manner that will lead to a meaningful outcome and allow testing the stated hypotheses?

Reviewer #1: Yes

Reviewer #2: Yes

3. Is the methodology feasible and described in sufficient detail to allow the work to be replicable?

Reviewer #1: Yes

Reviewer #2: Yes

4. Have the authors described where all data underlying the findings will be made available when the study is complete?

Reviewer #1: Yes

Reviewer #2: Yes

5. Is the manuscript presented in an intelligible fashion and written in standard English?

Reviewer #1: Yes

Reviewer #2: Yes

6. Review Comments to the Author

Reviewer #1: The authors have implemented all the suggestions provided. The manuscript has been now modified with all the required details.

Reviewer #2: Yes authors have addressed a few of the reviewer's comments. The major portion of the paper talks about the general introduction and review process. Statistical analysis of the outcome or any recommendation to future researchers is not elaborated.

The required improvements were applied in the introduction and discussion sections. 

- Few grammatical/typos can be looked into carefully.

The whole paper was double-checked for grammatical/typos. Thanks for the comment. 

The manuscript is really improved. 

Thanks so much for the constructive comments.

---

## [Editor Report · Decision Letter 2]

27 Dec 2022

PONE-D-22-10855R2A protocol for a systematic review of electronic early warning/track-and-trigger systems (EW/TTS) to predict clinical deterioration: Focus on automated features, technologies, and algorithms

PLOS ONE

Dear Dr. Rostam Niakan Kalhori,

Thank you for submitting your manuscript to PLOS ONE. After careful consideration, we feel that it has merit but does not fully meet PLOS ONE’s publication criteria as it currently stands. Therefore, we invite you to submit a revised version of the manuscript that addresses the points raised during the review process.

Statistical analysis part need to be explained more for better understandability of readers. 

We look forward to receiving your revised manuscript.

Kind regards,

Mohamed Yacin Sikkandar

Academic Editor

PLOS ONE

Journal Requirements:

Additional Editor Comments:

Statistical analysis Statistical analysis part need to be explained more for better understandability of readers.

Language can be further improved.

Referencing style can be formatted as per journal requirement.
---

## [Author Response · Author response to Decision Letter 2]

4 Jan 2023

Rebuttal Letter Round 3

Dear Editor of PLOS ONE

Thanks so much for the valuable comments on the manuscript (PONE-D-22-10855R1) entitled ''A protocol for a systematic review of early warning/track-and-trigger systems (EW/TTS) to predict clinical deterioration: Focus on automated features, technologies, and algorithms'' provided by reviewers. On behalf of the team of authors, I am writing the response to those comments which need reaction and correction. The changes are explained here in blue and presented as track changes in the text of the manuscript too. Furthermore, the revised clean version without track changes or highlights is submitted. 

Journal Requirements: 

The whole references were rechecked and sorted based on the Journals format. 

Additional Editor Comments:

1. Statistical analysis part needs to be explained more for better understandability of readers.

It is written with more explanation in more understandable version. 

2. Language can be further improved.

The whole doc was rechecked and the text was improved. 

3. Referencing style can be formatted as per journal requirements.

It is done. 

Thanks very much.

---

## [Editor Report · Decision Letter 3]

1 Mar 2023

A protocol for a systematic review of electronic early warning/track-and-trigger systems (EW/TTS) to predict clinical deterioration: Focus on automated features, technologies, and algorithms

PONE-D-22-10855R3

Dear Dr. Rostam Niakan Kalhori,

We’re pleased to inform you that your manuscript has been judged scientifically suitable for publication and will be formally accepted for publication once it meets all outstanding technical requirements.

Kind regards,

Mohamed Yacin Sikkandar

Academic Editor

PLOS ONE
---

## [Editor Report · Acceptance letter]

6 Mar 2023

PONE-D-22-10855R3 

A protocol for a systematic review of electronic early warning/track-and-trigger systems (EW/TTS) to predict clinical deterioration: Focus on automated features, technologies, and algorithms 

Dear Dr. Rostam Niakan Kalhori:

I'm pleased to inform you that your manuscript has been deemed suitable for publication in PLOS ONE. Congratulations! Your manuscript is now with our production department. 

Kind regards, 

on behalf of

Dr. Mohamed Yacin Sikkandar 

Academic Editor

PLOS ONE